# Genome-Wide Identification and Expression Analysis under Abiotic Stress of *BrAHL* Genes in *Brassica rapa*

**DOI:** 10.3390/ijms241512447

**Published:** 2023-08-04

**Authors:** Xiaoyu Zhang, Jiali Li, Yunyun Cao, Jiabao Huang, Qiaohong Duan

**Affiliations:** College of Horticulture Science and Engineering, Shandong Agricultural University, Tai’an 271018, China; zxy286110985@hotmail.com (X.Z.); gali20010917@163.com (J.L.); janecyy@163.com (Y.C.)

**Keywords:** *Brassica rapa*, *AHL*, bioinformatics, expression profile, abiotic stress

## Abstract

The AT-hook motif nuclear localized (*AHL*) gene family is a highly conserved transcription factor critical for the growth, development, and stress tolerance of plants. However, the function of the *AHL* gene family in *Brassica rapa* (*B. rapa*) remains unclear. In this study, 42 *AHL* family members were identified from the *B. rapa* genome and mapped to nine *B. rapa* chromosomes. Two clades have formed in the evolution of the *AHL* gene family. The results showed that most products encoded by *AHL* family genes are located in the nucleus. Gene duplication was common and expanded the *BrAHL* gene family. According to the analysis of *cis*-regulatory elements, the genes interact with stress responses (osmotic, cold, and heavy metal stress), major hormones (abscisic acid), and light responses. In addition, the expression profiles revealed that *BrAHL* genes are widely expressed in different tissues. *BrAHL16* was upregulated at 4 h under drought stress, highly expressed under cadmium conditions, and downregulated in response to cold conditions. *BrAHL02* and *BrAHL24* were upregulated at the initial time point and peaked at 12 h under cold and cadmium stress, respectively. Notably, the interactions between *AHL* genes and proteins under drought, cold, and heavy metal stresses were observed when predicting the protein-protein interaction network.

## 1. Introduction

The AT-Hook motif nuclear localized (*AHL*) gene family is a highly conserved transcription factor critical for the growth, development, and stress tolerance of plants [1,2,3,4]. AHL proteins consist of the AT-hook motif and the Plant and Prokaryote Conserved (PPC/DUF296) domain [5]. The AT-hook motif has a conserved palindromic core sequence (i.e., Arg-Gly-Arg) that binds to AT-rich DNA [6,7]. In terrestrial plants, the PPC/DUF296 domain is localized at the carboxyl end relative to the AT-hook motif [8]. The *AHL* gene interacts with other proteins through PPC/DUF296 to participate in plant biological activities [5]. It has been identified in terrestrial plant species such as *Arabidopsis thaliana* (*A. thaliana*) and *Oryza sativa* (*O. sativa*) [3,4].

*AHL* genes are implicated in the process in response to abiotic and biotic stresses. The drought avoidance and tolerance of *O. sativa* are positively affected by *OsAHL1* containing an AT-hook motif. Specifically, the overexpression of *OsAHL1* improves stress tolerance in rice [4]. In *Populus trichocarpa*, *PtrAHL34* induced by drought stress in roots and leaves can enhance drought tolerance [9]. *PtrAHL14* and *PtrAHL17* have a positive effect under cold conditions by modulating *PtrA/NINV7-*mediated Suc catabolism [10]. In *A. thaliana*, *AtAHL10* phosphorylation regulated by abscisic acid (ABA) induced 1 (*HAI1*) is involved in drought stress and defense processes [11]. The phosphorylation modifies the *AtAHL13* protein in response to pathogen-associated molecular pattern-triggered immunity [12]. The overexpression of *AtAHL20* is highly susceptible to Pseudomonas syringae and negatively affects plant immunity [13]. In *Cicer arietinum* L., *CaAHL18* exhibits a high expression level against the severe foliar disease Ascochyta blight [14]. In *Vernicia fordii*, *AHL* genes modulate *V. fordii* seed development and respond to Fusarium wilt, which exerts a negative effect on seed oil accumulation [15]. Based on previous studies, *AHL* genes are also critical in the physiological and developmental processes of plants. In *A. thaliana*, *AtAHL22* is involved in FLOWERING LOCUS T chromatin modification and delays the flowering time [16,17]. *AtAHL15* can form the secondary xylem. On the other hand, mutant *AHL15* reduces the development of secondary xylem [18]. *AtAHL18* modulates the architecture of the root system, and the mutant *AHL18* is shown to possess reduced primary root elongation and a lateral root number [19]. In *Zea mays*, barren stalk fastigiate-1 is an AT-hook protein forming the seed-bearing inflorescences and is also annotated as the maize ear [20].

*Brassica rapa (B. rapa,* Chinese cabbage) is a leafy vegetable that belongs to the cruciferous family. Notably, *B. rapa* experiences yield reduction under adverse conditions, including drought, extreme temperatures, and heavy metal stress [21]. Previous studies have demonstrated that *AHL* genes can mediate stress tolerance in different plants [8,9,10,11]. However, few studies elucidated the function of *AHL* genes in *B. rapa*. In this study, *BrAHL* genes were identified from the *B. rapa* genome. Their physicochemical properties, structures, and expression profiles were analyzed. It is hypothesized that the *BrAHL* genes in plants can respond to abiotic stress.

## 2. Results

### 2.1. Identification and Physicochemical Properties of the BrAHL Gene Family

In order to analyze the basic characteristics of the *BrAHL* gene family, 42 conserved *AHL* family members were identified from the whole genome and labeled as *BrAHL1* to *BrAHL42*. Their physical and chemical properties were analyzed by the ExPASy tool. The measured relative molecular weights ranged between 27,045.57 and 45,472.21 D, the protein isoelectric points ranged between 5.04 and 10.42, and the length of the amino acids ranged between 261 and 429 aa. Furthermore, more than 60% of the proteins had isoelectric points greater than 7, indicating that most *BrAHL* proteins were enriched in basic amino acids. Subcellular localization showed that over 90% of *AHL* proteins were located in the nucleus, suggesting the possibility of the *BrAHL* gene family being involved in intranuclear regulation, consistent with a previous finding that *AHLs* are nuclear-localized proteins [3]. *BrAHL10* is located in chloroplasts, and only *BrAHL27* is distributed in both the nucleus and cytosol. In addition, *BrAHL31*, *BrAHL32*, *BrAHL33*, *BrAHL34*, and *BrAHL35* can be detected in both the nucleus and chloroplasts (Appendix A).

### 2.2. Phylogenetic Analysis of the BrAHL Gene Family

A phylogenetic analysis was performed on the identified *AHL* family members to further investigate the evolutionary relationship of the *BrAHL* gene family. In Figure 1, branches indicate different evolutionary clades, with different colors representing different species. The number in each branch denotes the percentage of reliability. The selected species include *B. rapa*, *A. thaliana*, and *Solanum lycopersicum*, and the results indicate that two clades (A and B) are formed during the evolution of this gene family. All family members are unevenly distributed in two clades. Clade A consists of 13 *BrAHLs*, 15 *AtAHLs*, and 13 *SIAHLs*, and Clade B includes 29 *BrAHLs*, 14 *AtAHLs*, and 19 *SlAHLs*. The close phylogenetic relationship between *B. rapa* and the rest of the *AHL* families reveals the consistent evolution of *AHL* genes among different species, suggesting that the homologous genes may have similar functions.

### 2.3. Gene Structure and Conserved Motifs of the BrAHL Gene Family

The evolutionary relationship of gene families can be represented by the gene structure and motifs. In this study, the gene structure and motifs were explored by comparing coding sequences with corresponding genomic DNA sequences. In Figure 2B, the green boxes represent exons, the yellow boxes represent the conserved domain of *AHL* genes, and the black lines connecting exons denote introns. Most *BrAHL* genes in the same clade have similar exon intron lengths and numbers, especially the paralogous pairs in the same clade. It can be seen from Figure 2A that the number of exons in the *BrAHL* gene family is consistent (ranging from one to six), with most members containing five exons. All the members in Clade A have introns, while 12 members in Clade B have no intron. In addition, *BrAHL12* and *BrAHL16* have four introns, while *BrAHL42* has only one intron in Clade B (Figure 2B). The conserved motifs of 42 *BrAHL* members were analyzed, revealing the similarity of conserved motifs in the same clade. All the gene family members contain Motif 1 and Motif 5. Motif 6 only appears in Clade A, and Motif 9 appears only in Clade B. Most gene family members in Clade B have motif 7, while only five gene family members in Clade A have motif 7 (Figure 2C). Motif 5 and motif 8 consist of highly conserved R-G-R-P-R-K-Y and R-G-R-P amino acid sequences, respectively. Among them, R-G-R-P is the core sequence shared by the two motifs. It can prevent changes in DNA conformation by combining with the minor groove of DNA, resulting in the binding between transcription factors and the major groove [7] (Figure 2D). This result is consistent with the previous study on *Brassica napus* [22].

### 2.4. Collinear and Homologous Gene Pairs in the BrAHL Gene Family

To further investigate the evolutionary relationships, a homology map of the *AHL* genes was constructed using the genes of *A. thaliana*, a model plant belonging to Brassicaceae. The red lines in Figure 3 represent the collinear gene pairs, and the green blocks represent chromosomes. All the *BrAHL* family members share homologs with *A. thaliana*, indicating a good homology between the two species and the similar functions of the genes. According to the location of the *AHL* genes throughout the chromosomes, forty-two *BrAHL* genes are unevenly distributed on nine chromosomes, with one–seven genes on each chromosome. Chromosome A 09 contains the most genes with seven family members. Chromosomes A 01 and A 04 have six family members. Chromosomes A 03, A 06, and A 07 have five members. Chromosome A 05 has the least genes, with only one family member (Figure 3).

The collinearity of the *BrAHL* gene family was analyzed to select duplicate genes based on two criteria (comparison rate of two genes > 75% and comparative similarity > 75%). Groups of gene pairs located on different chromosomes were obtained. Five or fewer genes positioned within 100 kb on the same chromosome are considered tandem duplication [23]. *Ka* and *Ks* represent the substitutions at each synonymous and nonsynonymous site, respectively. The *Ka/Ks* values of most gene pairs are lower than 1, indicating that purifying selection affects the evolution of most gene pairs and suppresses the differentiation of duplicate genes. Only the *BrAHL27-BrAHL26* obtained from tandem duplication have *Ka/Ks* greater than 1, belonging to positive selection. In addition, most of the genes in the *BrAHL* gene family are characterized by segmental duplications, indicating a high degree of homology in this gene family (Appendix A).

### 2.5. Cis-Regulatory Element Analyses of the BrAHL Gene Family

*Cis*-regulatory elements affect the initiation and efficiency of gene transcription by binding to transcription factors [24]. In this study, *cis*-regulatory elements of the *BrAHL* gene family were analyzed to determine the potential functions of *BrAHL* genes, and a 2000-bp sequence upstream of the gene start codon was downloaded. Based on the promoter sequences of *BrAHL* genes, 20 types of *cis*-regulatory elements were predicted. Most of these elements were associated with abiotic stress responses, plant growth and development, and hormonal responses. *BrAHL01*, *BrAHL04*, *BrAHL06*, *BrAHL07*, *BrAHL08*, *BrAHL16*, *BrAHL18*, *BrAHL25*, and *BrAHL40* have drought-inducible *cis*-regulatory elements. *BrAHL02*, *BrAHL07*, *BrAHL08*, *BrAHL10*, *BrAHL13*, *BrAHL15*, *BrAHL16*, *BrAHL19*, *BrAHL21*, *BrAHL24*, *BrAHL34*, and *BrAHL39* have low-temperature responsiveness *cis*-regulatory elements. The number and distribution of promoter *cis*-regulatory elements vary significantly (Figure 4A). All 42 *BrAHL* genes contain considerable light-responsive elements, suggesting that the *BrAHL* genes may function in counteracting heavy metal stress. More than 90% of genes have anaerobic-responsive elements and five plant hormone-responsive elements. In addition, some members have elements related to stress responses, including drought and low temperatures (Figure 4B). Based on the above results, it can be speculated that the *BrAHL* genes may be involved in the regulation of stress, light, and hormone responses.

### 2.6. Expression Profile of BrAHL Genes in Different Tissues

In order to investigate the expression of *BrAHL* genes, the tissue expression profiles of this gene family in *B. rapa* root, stem, leaf, flower, silique, and callus were explored. The results revealed that each member differed in tissue expression. Therefore, it is speculated that the expression of *BrAHL* genes has different effects on plant development at different stages. As shown in Figure 5, *BrAHL02*, *BrAHL16*, *BrAHL17*, and *BrAHL20* are widely expressed in tissues, and *BrAHL04*, *BrAHL05*, *BrAHL08*, *BrAHL12*, *BrAHL19*, and *BrAHL29* exhibit a relatively low expression in different tissues. In contrast, *BrAHL13*, *BrAHL23*, *BrAHL25*, *BrAHL26*, *BrAHL27*, and *BrAHL35* are less expressed in organs. The high expressions of *BrAHL01*, *BrAHL02*, *BrAHL04*, *BrAHL16*, *BrAHL17*, *BrAHL18*, *BrAHL24*, *BrAHL28*, *BrAHL29*, and *BrAHL41* in roots suggest that they might participate in the regulation of drought stress. Notably, *BrAHL18* shows a higher expression in the leaf than in other tissues, and *BrAHL36* exhibits a higher expression level in silique than in other organs.

### 2.7. Expression Profile of BrAHL Genes under Osmotic Treatment

Drought stress negatively affects plant growth and development, decreasing yield and vegetable quality [25]. Under drought conditions, the root system is the primary organ that responds to stress [26]. To further investigate the effects of *BrAHL* genes under drought conditions, the RNA of *BrAHL* genes was extracted at 2, 4, 6, and 12 h of osmotic treatment, and the qRT-PCR was used to analyze the expression level of the ten family genes abundantly expressed in the roots. Half of the *BrAHL* family members were upregulated under osmotic treatment. As shown in Figure 6, *BrAHL01* is upregulated at all time points and peaks at 4 h. *BrAHL28* shows a similar trend and reaches the maximum at 6 h. Notably, *BrAHL01*, *BrAHL16*, *BrAHL17*, and *BrAHL41* are induced at 4 h of drought stress and are significantly suppressed at the other three time points, suggesting their potential effects on stress tolerance. *BrAHL02* shows moderate upregulations at all time points. *BrAHL18* is upregulated at 2 h and downregulated at 4 h, while *BrAHL04* exhibits the opposite trend.

### 2.8. Expression Profile of BrAHLs under Cold Treatment

Low temperatures can suppress plant growth and food quality, potentially generating reactive oxygen species (ROS) [27,28]. To have insights into the expression of *BrAHL* genes under such stress, RNA samples of the family genes under cold treatment at 2, 4, 6, and 12 h were extracted, and the expression levels of ten genes in the *BrAHL* gene family were analyzed using qRT-PCR. The presence of four cold stress-induced genes indicated their involvement in regulating biological processes under low temperatures. *BrAHL02* was upregulated at the first two time points, was significantly downregulated at 6 h, and peaked at 12 h. *BrAHL18* was elevated gradually at all durations and reached the climax at 12 h. *BrAHL24* was downregulated at 2 h and 6 h and dramatically upregulated at 4 h and 12 h. The expression trend of *BrAHL28* was the opposite. It is worth noting that *BrAHL01*, *BrAHL04*, *BrAHL16*, *BrAHL17*, *BrAHL29* and *BrAHL41* were all inhibited at tested periods while coping with cold stress (Figure 7).

### 2.9. Expression Profile of BrAHL Genes under Cadmium (Cd) Treatment

Cd is a highly toxic heavy metal that substantially risks plant growth [29]. To better explore the expression of *BrAHL* genes under Cd stress, the RNA of the gene family under Cd treatment at 2, 4, 6, and 12 h was extracted, and the expression level of ten genes in the *BrAHL* gene family was analyzed using qRT-PCR. The results showed that most family genes were upregulated, indicating their positive response to Cd exposure. *BrAHL01*, *BrAHL04*, *BrAHL16*, *BrAHL17*, and *BrAHL41* showed a consistent trend and peaked at 6 h under Cd stress. *BrAHL02*, *BrAHL28*, and *BrAHL29* were suppressed at all time points. *BrAHL18* and *BrAHL24* were slightly upregulated at the first time point and suppressed more significantly than the control at the next three time points (Figure 8).

### 2.10. Protein-Protein Interaction Networks of the AHL Genes

Protein-protein interaction networks are composed of proteins involved in basic life processes through mutual interactions, such as gene expression regulation, signal transduction, and metabolism control. The protein-protein interaction networks were predicted, whereby circles represent the different gene members, and lines denote the interactions. In addition, the degree centrality of nodes exhibits a positive correlation with the circle sizes and shades of color. Given that *BrAHL* genes are closely related to *AtAHL* genes, protein interactions were predicted to reveal the potential functions of these proteins. *AT2G33620* (homologous gene of *BrAHL16*) was observed to interact with HAI1, which functions as a negative regulator in response to osmotic stress and drought [11]. It is inferred that *BrAHL16* is probably involved in regulation under drought stress (Figure 9A). *AT4G22770* (homologous gene of *BrAHL02*) interacted with AT-HSFA5, a heat shock protein (HSP). HSPs are induced by unfavorable environments, suggesting that *BrAHL02* potentially affects the process under cold conditions (Figure 9B). *AT3G04590* (homologous gene of *BrAHL24*) was found to have interactions with far-red elongated hypocotyls (FRSs) in regulating the light control of plant development. Notably, photosynthesis is the primary process affected by stress conditions, e.g., heavy metal stress [30,31,32]. It can be speculated that *BrAHL24* participates in heavy metal stress tolerance (Figure 9C).

## 3. Discussion

*AHL* genes are implicated in plant growth, physiological processes, and stress tolerance. The role of *AHL* genes was not recognized in *B. rapa*. In this study, 42 *BrAHL* genes were identified in *B. rapa,* and their genomic features were analyzed, such as evolutionary relationships, gene structures, conserved motifs, duplicate gene pairs, *cis*-regulatory elements, and expression profiles.

Gene duplication serves as a main driving force in the evolution of genomes and genetic systems [33,34], facilitating the formation of novel gene functions and species evolution [35,36]. Duplicate genes provide raw materials for new genes, which in turn promotes the generation of new functions. Four major patterns of evolution involve fragmental, tandem, whole genome, and gene transposition duplications [37,38]. Among them, segmental and tandem duplications are considered primary contributors to the expansion of plant gene families [39]. Tandem duplication derives from unequal crossing-over events [40], and multiple episodes of these crossovers may increase or decrease the copy number in different gene families. Tandem replication mainly occurs in the region of chromosome recombination and forms a gene cluster with identical sequences and functions. Segmental duplication results in the duplicated blocks of genomic DNA typically within 200 kb in size, and these segments contain high-copy repeats and gene sequences characterized by an intron exon structure [41]. Brassicaceae genomes were estimated to undergo three rounds of whole genome duplication [42,43]. In the *BrAHL* gene family, most *Ka/Ks* values of studied gene pairs were lower than 1, indicating that purifying the selection influences the evolution of most gene pairs and suppresses the differentiation of duplicate genes. Remarkably, *BrAHL27* and *BrAHL26* had *Ka/Ks* values greater than 1, indicative of positive selection. In the case of *BrAHL* genes, segmental duplication was primarily responsible for the gene family expansion, and one instance of tandem duplication was also seen. The amplification of genes by tandem duplication was more closely related to abiotic and biotic stress than non-tandem duplications [25,26,44]. In this sense, it is inferred that gene duplication is involved in *BrAHL* gene family expansion and plant stress tolerance.

Drought stress often negatively influences plant growth and development, potentially decreasing yield and vegetable quality [45]. Under drought conditions, roots are the primary organs coping with stress [46]. Given this scenario, ABA can be employed to regulate the physiological activities of plants under drought stress [27,47]. Notably, the MYB binding site (*MBS*) *cis*-regulatory element is involved in drought-inducibility elements. An ABA-responsive element is involved in ABA responsiveness and plays an important role in dealing with drought stress. In the *BrAHL* gene family, *BrAHL16* expressed abundantly in roots and possessed *MBS* and ABA-responsive elements. This gene was upregulated at 4 h under osmotic stress and downregulated at the following time points. In *O. sativa*, *OsAHL1* was induced under drought stress. In *OsAHL1* over-expressing plants, water loss was slower, and the volume of the total root, upper root, and lower root in plants was higher than in wild-type plants under drought stress. *OsAHL1* regulated its target genes (*HSP101*, *OsCDPK7*, *OsRNS4*, *Rab16b*, and *AP2-EREBP*) to improve drought tolerance. It is indicated that drought resistance is enhanced by the over-expression of *OsAHL1* [4]. In *A. thaliana*, some *AtAHL10* phosphorylation sites were de-phosphorylated by *HAI1*, ultimately suppressing plant growth under drought stress. Notably, *AHL10* had more seedling weight and root elongation than the wild type under moderate soil drying. *HAI1*-regulated phosphorylation of *AtAHL10* could affect plant growth under drought stress [11]. In *Populus trichocarpa*, *PtrAHL34* with an *MBS cis*-element exhibited a high expression level in roots and was induced by drought stress at 6 h and declined to a normal level at 24 h [9]. It can be elucidated that *BrAHL16* is involved in the regulation of drought stress.

Cold stress hampers plant growth and food quality and may produce ROS [28,48]. The response of plants to extreme temperatures is mediated by plant hormones [49,50]. One fundamental mechanism is to induce ROS production and activate nicotinamide adenine dinucleotide phosphate oxidases in response to temperature changes [51,52]. A ong-terminal repeat *cis*-regulatory element is vital in low-temperature response. In the *BrAHL* gene family, *BrAHL02* had low-temperature *cis*-regulatory elements and was gradually expressed at 2 h and 4 h, was downregulated at 6 h, and peaked at 12 h. From the protein-protein interaction networks, the homologous gene in *A. thaliana* (*AT4G22770*) interacted with AT-HSFA5. HSFA4 and HSFA5 belong to distinct subgroups within the class A HSF, characterized by identical DNA-binding domains and conserved C-terminal motifs [53]. Heat shock factors (HSFs) are transcriptional regulators that mediate adverse conditions such as extreme temperatures [54,55]. Some HSFs act as molecular peroxide sensors that respond to alterations in ROS levels under stress [56]. HSFA4A transcription is stimulated by stress conditions (e.g., cold stress) and produces ROS [29]. Based on the findings, *BrAHL02* probably plays an important role under cold conditions.

Cd exposure is considered a potential threat to plant growth due to its toxic effects. [30]. Notably, photosynthesis is the primary process affected by adverse conditions such as heavy metal stress [31,32,57]. Photosynthetic response to stress conditions is extremely complicated and occurs at different sites of cells or leaves at various time scales [58]. In the *BrAHL* gene family, *BrAHL24* with light-responsive *cis*-regulatory elements was induced at the initial two periods, was downregulated at 6 h, and peaked at 12 h. Considering protein-protein interaction networks, the homologous gene *AT3G04590* interacted with far-red elongated hypocotyls (FRSs). FRSs are crucial in the light control process that regulates the development of roots and flowers [59]. It is concluded that *BrAHL24* probably has potential functions under Cd stress.

## 4. Materials and Methods

### 4.1. Identification of AHL Members and Their Physical and Chemical Properties

A total of 29 *A. thaliana* AHL protein sequences were downloaded from the TAIR database (http://www.arabidopsis.org/, accessed on 1 July 2023). The possible members of the *BrAHL* family were searched through the two-way BLAST in the *B. rapa* genome version 1.5 (http://brassicadb.cn/, accessed on 1 July 2023) [60]. The gene family was further identified using conserved domain verification [61]. The physicochemical properties, such as the isoelectric point and molecular weight, were analyzed on the ExPASy server (https://www.expasy.org/, accessed on 1 July 2023) [62]. The data were acquired in batches using TBtools (version 1.120) and sorted out [63]. The subcellular localization was predicted using WoLF PSORT (http://www.genscript.com/wolf-psort.html, accessed on 1 July 2023) [64].

### 4.2. Phylogenetic Analysis of the Family Genes

According to the ID information of the identified family genes, the protein sequences were combined with other species utilizing ClustalW [65] multiple sequence alignment. The reliability of the phylogenetic tree was evaluated using 1000 bootstrap duplications. At the same time, based on Jones Taylor Thornton (JTT) model and gamma distribution, we also constructed the phylogenetic tree using the maximum-likelihood (ML) method in MEGA-11 [45]. The evolutionary tree was beautified on the website (https://itol.embl.de/itol.cgi, accessed on 1 July 2023) [66].

### 4.3. Analysis of Gene Structure and Cis-regulatory Elements

MEME tool (http://meme-suite.org/, accessed on 1 July 2023) and Batch CD-Search on the NCBI website were adopted to predict conserved motifs and domains, respectively [67]. TBtools software (version 1.120) was employed to combine the conserved motif and domain with the phylogenetic tree for mapping [63]. The 2000-bp sequence upstream of the *BrAHL* gene initiation codon was downloaded through the Ensembl Plants website (http://plants.ensembl.org/index.html, accessed on 1 July 2023) [60].

### 4.4. Analysis of Collinearity and Gene Pairs

The MCScanX [23] was utilized to identify the collinear relationships between the chromosomes within the genome. Sequentially, duplication analysis was performed on the identified duplicate gene pairs between family genes. The results were illustrated using the Circos software [68]. Synonymous and nonsynonymous substitutions were selected to calculate the *Ka/Ks* values for all paralogous genes [69].

### 4.5. Analysis of the Expression Profiles

RNA-seq data (accession number: GSE43245) were collected for the transcriptome sequencing of *B. rapa* tissues (http://brassicadb.cn, accessed on 1 July 2023). All transcriptomic expression data were homogenized using log_2_. A cluster heatmap of the gene family expression was constructed to show the differential expression of the family members in different tissues [70].

### 4.6. Total RNA Extraction and qRT-PCR

Chinese cabbage seedlings were grown to six leaves in Hoagland nutrient solution (Coolaber, Beijing, China) and treated with PEG 6000 (15%), 4 °C, and 100 µM CdCl_2_ to simulate osmotic, cold, and Cd stress in a hydroponic system for 2, 4, 6 and 12 h. Normal Chinese cabbage seedlings were used as a control. Following RNA extraction, RNA molecules in the sample were verified with 1% agarose gel electrophoresis, and retrotranscription was performed to prepare the cDNA. The qRT-PCR primers were obtained from the qPrimerDB-qPCR primer database (https://biodb.swu.edu.cn/qprimerdb/, accessed on 1 July 2023). *BrActin2* was used as the reference gene and synthesized by Qingdao Qingke Zixi Biotechnology Co., Ltd. (Qingdao, China). The total RNA sample was obtained using an RNA extraction kit (Black, Beijing, China) using HiScript II QRT SuperMix for qPCR. The resulting RNA was then reverse transcribed using a total RNA extraction kit (Novizan, Nanjing, China) to obtain the cDNA. Ten-fold dilution with dd H_2_O was added to test the expression of *BrAHL* family genes. The 20 L reaction system was configured according to the following instructions of the ChamQ Universal SYBR qPCR Master Mix kit (Novizan, Nanjing, China): 10 μL of SYBR green master mix, 2 μL of cDNA, 7.2 μL of dd H_2_O, and 0.4 μL each of the upstream and downstream primers. The PCR reaction was performed with a qTOWER3 (Analytik Jena AG, Jena, Germany) at 40 cycles of 95 °C for 30 s, 95 °C for 10 s, and 60 °C for 22 s. The relative expression of the Chinese cabbage genes was calculated using the 2^−∆∆CT^ method [71] and presented by TBtools (version 1.120) [63]. The primer sequences are shown in Appendix A.

### 4.7. Protein-Protein Interaction Network Analysis

STRING (http://cn.string-db.org/, accessed on 1 July 2023) [27] is a comprehensive protein-protein interaction database that includes known and predicted interactions. The protein sequences of the studied family genes were submitted to this database to query the relationships among the genes and subsequently map the protein-protein interaction networks utilizing Cytoscape (version 3.9.1) [72].

## 5. Conclusions

In this study, 42 *AHL* family members were identified from the *B. rapa* genome and mapped to nine *B. rapa* chromosomes. The expression profile revealed that *BrAHLs* were widely expressed in different tissues. Specifically, *BrAHL16* was probably involved in drought stress response. *BrAHL02* was induced under cold stress. *BrAHL2*4 may undergo regulation under Cd conditions. Notably, the protein-protein interaction network prediction demonstrated that *AT2G33620* (homologous gene of *BrAHL16*) interacted with HAI1. It is speculated that *BrAHL16* participates in the regulation under drought stress. *AT4G22770* (homologous gene of *BrAHL02*) was revealed to interact with AT-HSFA5, suggesting that *BrAHL02* functions under cold conditions. *AT3G04590* (homologous gene of *BrAHL24*) had interactions with far-red elongated hypocotyls (FRSs), indicating *BrAHL24*'s involvement in response to heavy metal stress.

## Figures and Tables

**Figure 1 ijms-24-12447-f001:**
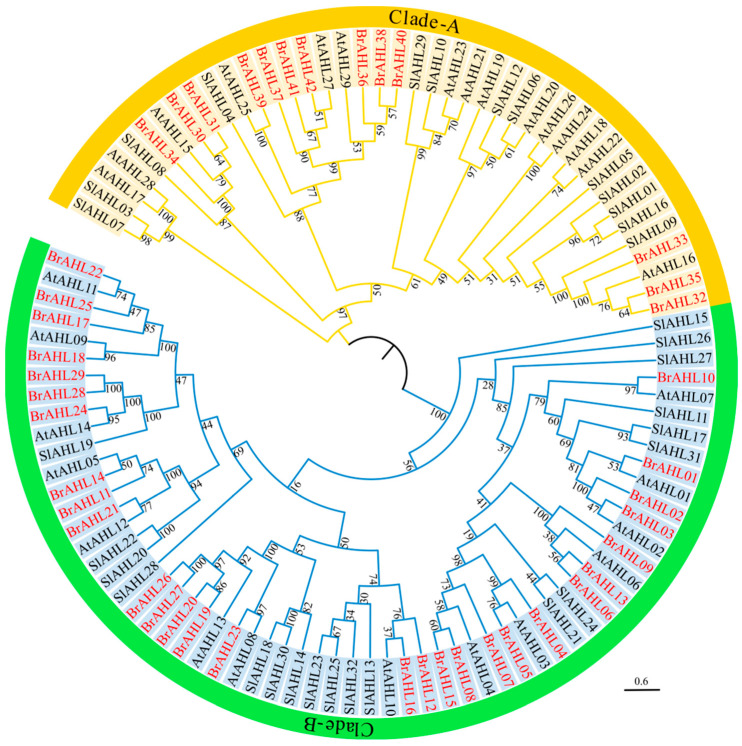
Phylogenetic analysis of the *AHL* gene family between *B. rapa* and *A. thaliana*, and *S. lycopersicum*. Branches indicate different evolutionary clades, with different colors representing different species. The number in each branch denotes the percentage of reliability.

**Figure 2 ijms-24-12447-f002:**
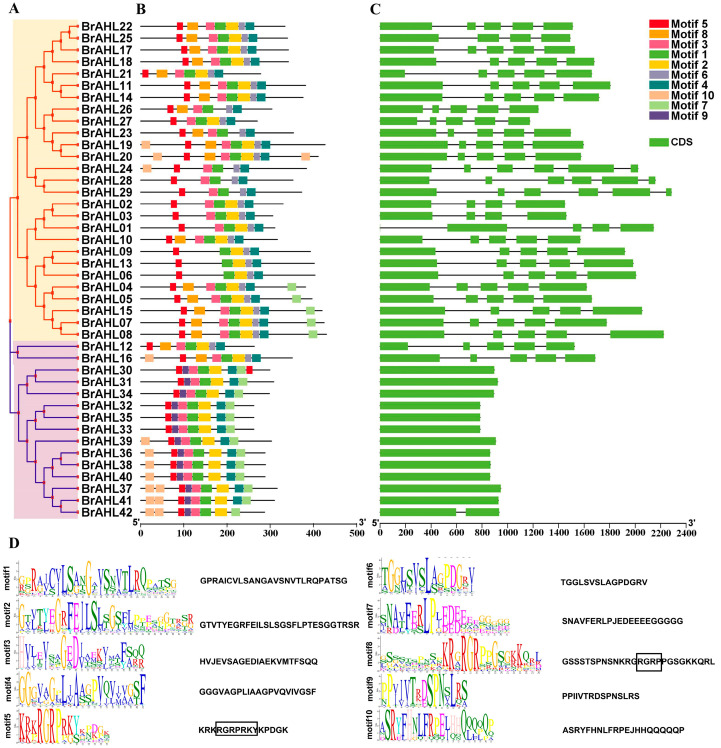
Analysis of the gene structure and protein conserved domains of *BrAHL* genes. (**A**). Phylogenetic analysis of *BrAHL* genes. (**B**). Gene structure of *BrAHL* genes. (**C**). Conserved protein motifs of *BrAHL* genes. (**D**). Motifs of *BrAHL* genes. The black boxes represent two amino acid sequences, respectively.

**Figure 3 ijms-24-12447-f003:**
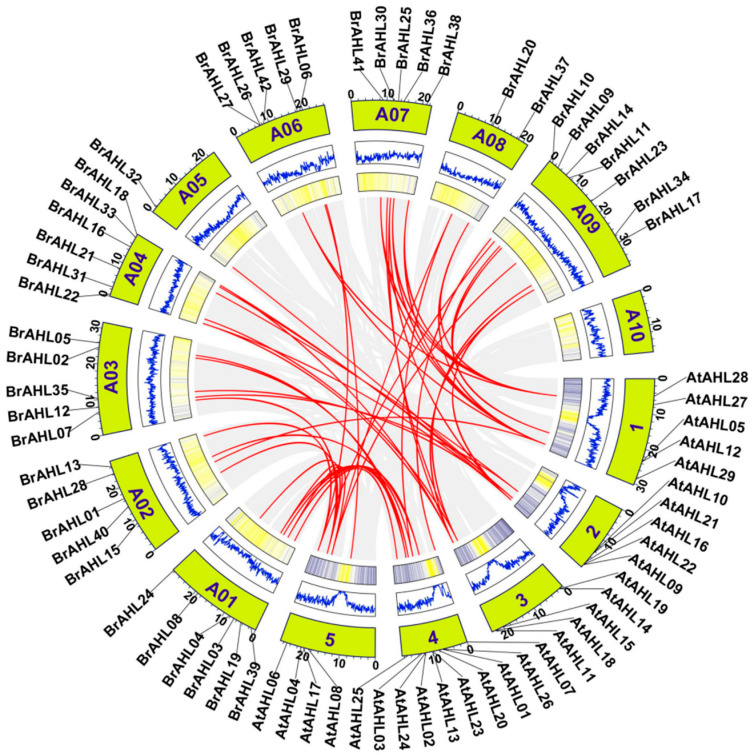
Collinearity between *B. rapa* and *A. thaliana*. The red lines represent the collinear gene pairs, and the green blocks represent chromosomes.

**Figure 4 ijms-24-12447-f004:**
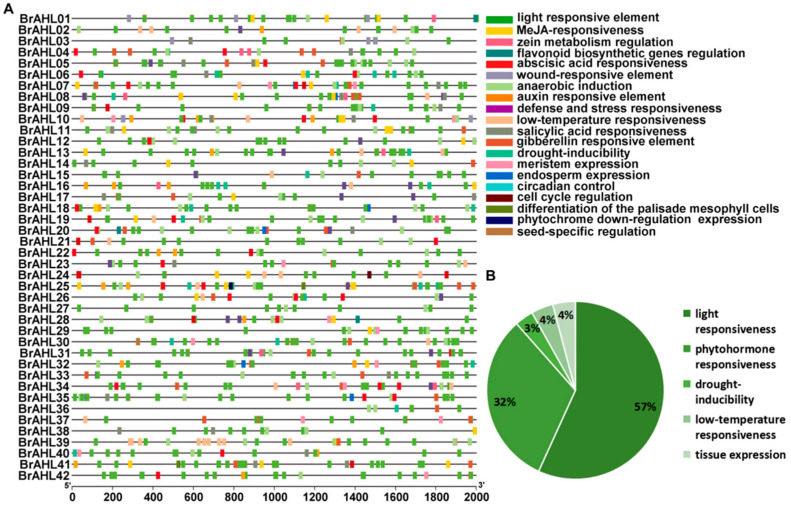
*Cis*-regulatory elements analysis in the promoter region of *BrAHL* genes. (**A**). *Cis*-regulatory elements of *BrAHL* genes. (**B**). Proportion of *cis*-regulatory elements.

**Figure 5 ijms-24-12447-f005:**
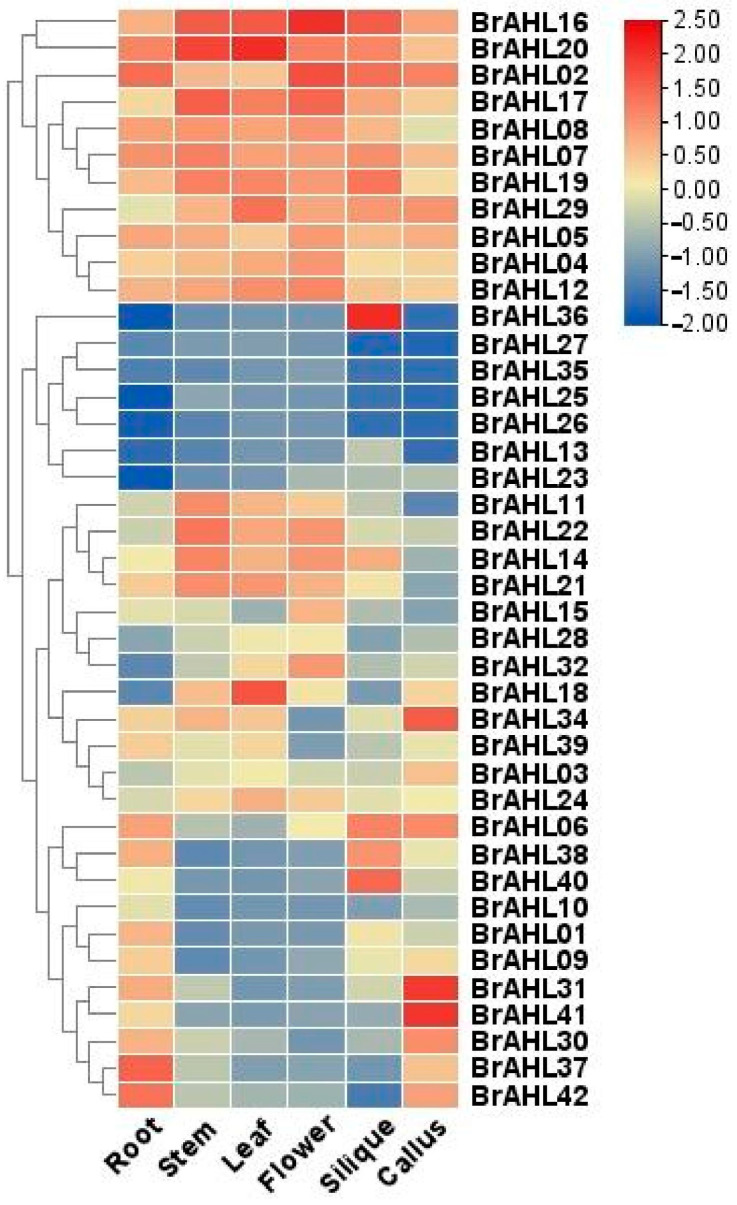
Expression of *BrAHL* genes in different tissues. Red color represents increased expression, and blue denotes decreased expression.

**Figure 6 ijms-24-12447-f006:**
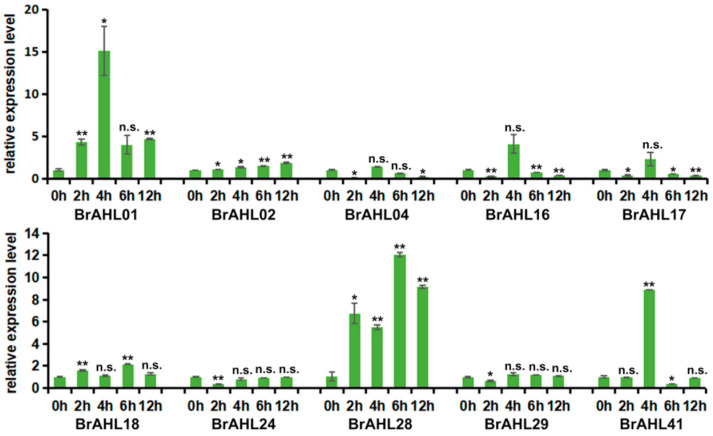
Expression of *BrAHL* genes under drought stress. Data are presented as means (±SD) of three biological replicates. Asterisks or n.s. above the data bars indicate a significant difference (two-tailed *t*-test * *p* < 0.05 ** *p* < 0.01) or no significant difference, respectively.

**Figure 7 ijms-24-12447-f007:**
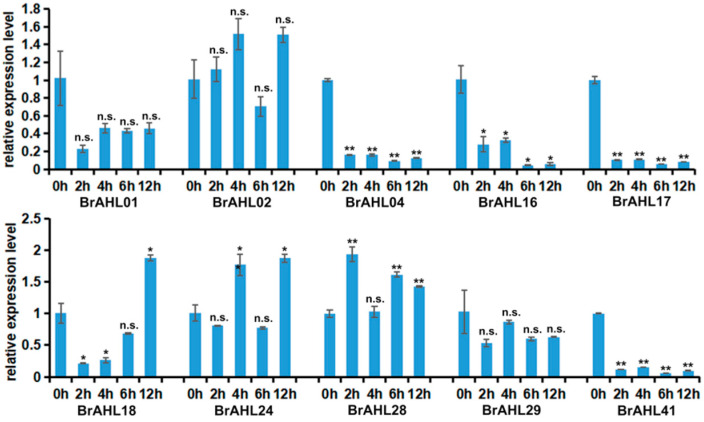
Expression of *BrAHL* genes under cold stress. Data are presented as means (±SD) of three biological replicates. Asterisks or n.s. above the data bars indicate a significant difference (two-tailed *t*-test * *p* < 0.05 ** *p* < 0.01) or no significant difference, respectively.

**Figure 8 ijms-24-12447-f008:**
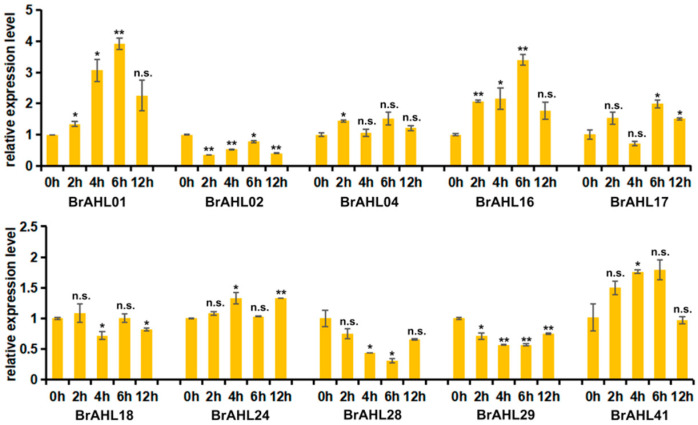
Expression of *BrAHL* genes under Cd stress. Data are presented as means (±SD) of three biological replicates. Asterisks or n.s. above the data bars indicate a significant difference (two-tailed *t*-test * *p* < 0.05 ** *p* < 0.01) or no significant difference, respectively.

**Figure 9 ijms-24-12447-f009:**
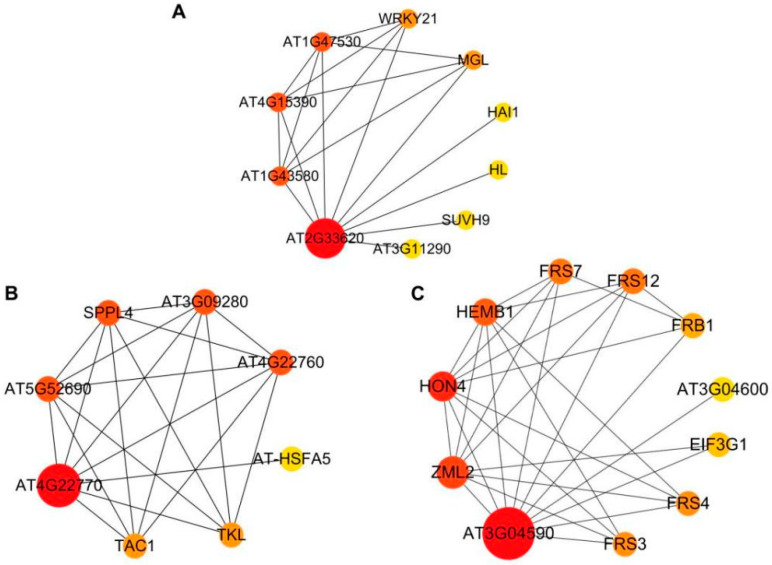
Protein-protein interaction network prediction of *AHL* genes. Circles represent the proteins. Lines represent the interaction. The degree centrality of nodes exhibits a positive correlation with the circle sizes and shades of color. (**A**). *AT2G33620* (homologous gene of *BrAHL16*) PPIs. (**B**). *AT4G22770* (homologous gene of *BrAHL02*) PPIs. (**C**). *AT3G04590* (homologous gene of *BrAHL24*) PPIs.

## Data Availability

Data are contained within the article/Appendix A.

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
