# Peer review of "Genome-Wide Identification and Expression Analysis under Abiotic Stress of *BrAHL* Genes in *Brassica rapa"

_ijms, 2023, doi:10.3390/ijms241512447_

Round 1

Reviewer 1 Report

Dear Authors/Editor,

I had an opportunity to review the manuscript entitled: “Genome-wide identification and expression analysis under abiotic stress of BrAHL genes in Brassica rapa” which is considered for publication in IJMS journal. The article itself present the fashionable checking the transcriptome under different the stress however in some parts the manuscript is short laconic not adequate prepared to the journal rules and even sometimes the presentation results is low quality to read exactly result. Article need improvement to fit to the IJMS standards. The issues which must be changed is present in a form of list below:

A.      Introduction section.

It has problem with thinking continuity it is separated on short different sections which creates some illogical effect. This section must be rewrite from the start in one logic text and what is the most important it need logically formulated aim or and hypothesis of research as IJMS journal rules stands for. Now the no aim could be logically driven from this part of manuscript.

B.      Results section

The description of results is extremely short in comparison to amount of results and concussions that authors presented. The results part must be enlarge or Authors must change this to communication (short communication) not full pledged article. The comparison of genes Figure 1-Figure 3 of Brassica rappa to Oryza sativa is completely illogical. The monocots has different structure of gene and expression patterns. Authors must perform analyses once more with elimination of any monocot in they analyses. Figure 2 is overloaded with data and low quality is extremely difficult to analyzes the results. Authors must split this figure to separate and add them in HD quality or better. Figure 3 again problem with monocots and it is too small and difficult to read. Figure 7,8 and 9 are extremely low quality any data could be observed the size of markings of statistical significance was unable to observe any significance. Authors must add 3 figures in much higher quality.

C.      Material and methods section.

In general is too laconic has extremely few citation with is extremely strange if authors used transcriptomic analyses which is well presented in current literature. The lack of citation suggest that authors developed new methods of analysis which simply is not true. The shorting’s and lack of citations creates the impression of ethical problems with this sections. Moreover many places methods ignore completely IJMS publication rules. Both elements must be changed.

D.      Comment to the level of preparation in general.

Authors must point by point check the IJMS publication rules and correct manuscript according it. The best evidence of sloppiness is Reference section in which on 59 positions any is not prepared according IJMS publication rules and some positions have even different size of fonts.

Sincerely,

Reviewer 2 Report

Dear Authors,

Reviewer comments ijms-2528992

The manuscript entitled „Genome-wide identification and expression analysis under abiotic stress of BrAHL genes in Brassica rapa“ provides a comprehensive study on BrAHL genes as important transcrition factors involved in plant stress responses in Chinese cabbage (Brassica rapa). The manuscript provides a phylogenetic analysis of BrAHL gene family together with AtAHL, SlAHL and OsAHL genes, structural and sequence motifs analysis, synteny analysis of BrAHL genes with respect to AtAHL and OsAHL genes. STRING tool was employed to predict protein-protein interactions (PPIs) for selected BrAHL genes. Furthemore, expression analysis of BrAHL genes with respect to 15% PEG-6000, 4 °C and 100 μM CdCl2 as osmotic, cold, and heavy metal stresses, respectively, for 2, 4, 6 and 12 h. I can recommend the present manuscript for publication in Intwernational Journal of Molecular Sciences. However, I have three major comments and several minor (formal) comments on the present manuscript below:

1/ Terminology: For 15% PEG-6000 treatment, the term „osmotic stress“ , not „drought stress“ has to be used.

2/ In Figure 1 showing the phylogenetic tree of BrAHL, AtAHL, SlAHL and OsAHL genes, appropriate data have to be added to the phylogenetic tree, i.e., a scale bar and numbers to each node indicating probability of each branching point per 1000 bootstrap replicates determined by Neighbour-Joining (N-J) method as described in Materials and methods.

3/ In Materials and methods, date of access has to be added to each database since the data in the databases should change during the time.

Minor (formal) comments on the text:

Abstract, line 15: Modify the statement as follows: „The analysis of cis-regulatory elements elucidated that the genes respond to stress (osmotic, cold, and heavy metal stress), major hormones (abscisic acid), and light.

Introduction, line 43: Modify the statement as follows: „AHLs are also implicated in response to biotic and abiotic stresses.“

Results, line 119: Replace the word „less“ with „lower“ in the statement „Most Ka/Ks values of gene pairs were lower than 1….“

Line 131: Modify the word form „drought-inducible cis-regulatory elements“ (not „drought-inducibility cis-regulatory elements“).

Results, Figure 5 legend, lines 157-158: Modify the words „higher expression“ and „lower expression“ to „incrteased expression“ and „decreased expression“, respectively.

Results, Figure 7 and Figure 8 legends: Add the word „respectively“ at the end of the statement „Asterisks or n.s. directly above the data bars indicate significant difference (two-tailed t test *P0.05 **P0.01) or no significant difference, respectively.“

Results, line 218: Add the word „tolerance“ at the end of the statement „It could eb speculated that BrAHL24 likely participates in heavy metal stress toelrance (Figure 9C).“

Discussion, line 263: Modify the statement as follows: „In OsAHL1 over-expressing plants,…“

Discussion, line 299: For „FRSs“ proteins, the full protein name or protein family name has to be given.

In Materials and methods, the date of access has to be given for TAIR database, STRING tool, and both the web address and date of access have to be given for B. rapa genome version 1.5 database.

Conclusion, line 372: For „FRSs“ proteins, the full protein name or protein family name has to be given.

Final recommendation: Accept after a minor revision.

Dear Authors,

Reviewer comments ijms-2528992

The manuscript entitled „Genome-wide identification and expression analysis under abiotic stress of BrAHL genes in Brassica rapa“ provides a comprehensive study on BrAHL genes as important transcrition factors involved in plant stress responses in Chinese cabbage (Brassica rapa). The manuscript provides a phylogenetic analysis of BrAHL gene family together with AtAHL, SlAHL and OsAHL genes, structural and sequence motifs analysis, synteny analysis of BrAHL genes with respect to AtAHL and OsAHL genes. STRING tool was employed to predict protein-protein interactions (PPIs) for selected BrAHL genes. Furthemore, expression analysis of BrAHL genes with respect to 15% PEG-6000, 4 °C and 100 μM CdCl2 as osmotic, cold, and heavy metal stresses, respectively, for 2, 4, 6 and 12 h. I can recommend the present manuscript for publication in Intwernational Journal of Molecular Sciences. However, I have three major comments and several minor (formal) comments on the present manuscript below:

1/ Terminology: For 15% PEG-6000 treatment, the term „osmotic stress“ , not „drought stress“ has to be used.

2/ In Figure 1 showing the phylogenetic tree of BrAHL, AtAHL, SlAHL and OsAHL genes, appropriate data have to be added to the phylogenetic tree, i.e., a scale bar and numbers to each node indicating probability of each branching point per 1000 bootstrap replicates determined by Neighbour-Joining (N-J) method as described in Materials and methods.

3/ In Materials and methods, date of access has to be added to each database since the data in the databases should change during the time.

Minor (formal) comments on the text:

Abstract, line 15: Modify the statement as follows: „The analysis of cis-regulatory elements elucidated that the genes respond to stress (osmotic, cold, and heavy metal stress), major hormones (abscisic acid), and light.

Introduction, line 43: Modify the statement as follows: „AHLs are also implicated in response to biotic and abiotic stresses.“

Results, line 119: Replace the word „less“ with „lower“ in the statement „Most Ka/Ks values of gene pairs were lower than 1….“

Line 131: Modify the word form „drought-inducible cis-regulatory elements“ (not „drought-inducibility cis-regulatory elements“).

Results, Figure 5 legend, lines 157-158: Modify the words „higher expression“ and „lower expression“ to „incrteased expression“ and „decreased expression“, respectively.

Results, Figure 7 and Figure 8 legends: Add the word „respectively“ at the end of the statement „Asterisks or n.s. directly above the data bars indicate significant difference (two-tailed t test *P0.05 **P0.01) or no significant difference, respectively.“

Results, line 218: Add the word „tolerance“ at the end of the statement „It could eb speculated that BrAHL24 likely participates in heavy metal stress toelrance (Figure 9C).“

Discussion, line 263: Modify the statement as follows: „In OsAHL1 over-expressing plants,…“

Discussion, line 299: For „FRSs“ proteins, the full protein name or protein family name has to be given.

In Materials and methods, the date of access has to be given for TAIR database, STRING tool, and both the web address and date of access have to be given for B. rapa genome version 1.5 database.

Conclusion, line 372: For „FRSs“ proteins, the full protein name or protein family name has to be given.

Final recommendation: Accept after a minor revision.

Round 2

Reviewer 1 Report

Dear Authors,

Most of my comments was take into consideration. But because references list is still needed to improved again any position is not according IJMS rules please check it again. I recommend minor revision which enables the corrections. Check the rules of the fonts, bold elements italics etc. in reference

Sincerely,
